# Change of human mobility during COVID-19: A United States case study

**Justin Elarde[1], Joon-Seok Kim[1], Hamdi Kavak[2], Andreas Züfle[1], Taylor Anderson[1]***

1 Department of Geography and Geoinformation Science, George Mason University, Fairfax, VA, United States of America, 2 Department of Computational and Data Sciences, George Mason University, Fairfax, VA, United States of America

\* tander6@gmu.edu

**Data Availability Statement:** The data underlying the results presented in the study are available from the SafeGraph Data for Academics (https://www.safegraph.com/academics). No cost access to SafeGraph data is available for academic non-

## Abstract

With the onset of COVID-19 and the resulting shelter in place guidelines combined with remote working practices, human mobility in 2020 has been dramatically impacted. Existing studies typically examine whether mobility in specific localities increases or decreases at specific points in time and relate these changes to certain pandemic and policy events. However, a more comprehensive analysis of mobility change over time is needed. In this paper, we study mobility change in the US through a five-step process using mobility footprint data. (Step 1) Propose the *Delta Time Spent in Public Places* ($\Delta TSPP$) as a measure to quantify daily changes in mobility for each US county from 2019-2020. (Step 2) Conduct Principal Component Analysis (PCA) to reduce the ΔTSPP time series of each county to lower-dimensional latent components of change in mobility. (Step 3) Conduct clustering analysis to find counties that exhibit similar latent components. (Step 4) Investigate local and global spatial autocorrelation for each component. (Step 5) Conduct correlation analysis to investigate how various population characteristics and behavior correlate with mobility patterns. Results show that by describing each county as a linear combination of the three latent components, we can explain 59% of the variation in mobility trends across all US counties. Specifically, change in mobility in 2020 for US counties can be explained as a combination of three latent components: 1) long-term reduction in mobility, 2) no change in mobility, and 3) short-term reduction in mobility. Furthermore, we find that US counties that are geographically close are more likely to exhibit a similar change in mobility. Finally, we observe significant correlations between the three latent components of mobility change and various population characteristics, including political leaning, population, COVID-19 cases and deaths, and unemployment. We find that our analysis provides a comprehensive understanding of mobility change in response to the COVID-19 pandemic.

## 1 Introduction

Human mobility plays a crucial role in spreading an infectious virus such as SARS-CoV-2 and has been instrumental in the onset of the COVID-19 pandemic. In response to this global

commercial work. Data derived and aggregated from the SafeGraph data is available at https://github.com/GMU-GGS-NSF-ABM-Research/Mobility-Trends. Please see the supplemental material for more details.

**Funding:** Award 1: T.A. A.Z. H.K. Award no: 2030685 Funder: National Science Foundation Funder website: nsf.gov. The funders had no role in study design, data collection and analysis, decision to publish, or preparation of the manuscript. Award 2: T.A. A.Z. H.K. Award no: 2109647 Funder: National Science Foundation Funder website: nsf. gov. The funders had no role in study design, data collection and analysis, decision to publish, or preparation of the manuscript.

**Competing interests:** The authors have declared that no competing interests exist.

crisis, non-pharmaceutical interventions (NPIs), including stay-at-home orders and social distancing guidelines, have been implemented [1], reducing physical contacts and resulting in significant changes to normal mobility patterns [2]. These changes can be observed using individual-level mobility data such as mobile phone data embedded with Bluetooth and global positioning system (GPS), collected actively through Call Detail Records (CDRs) and passively through the use of smartphone applications (apps).

Individual-level mobility data are typically anonymized and aggregated to various spatial resolutions to produce a range of different *mobility indicators* (see Table 1). As part of their COVID-19 Data Consortium efforts, SafeGraph Inc. [3] has made available a comprehensive set of mobility indicators ranging from mean distance traveled to median dwell time at and away from home for each census block group in the US. Descartes Labs Inc. [4] makes available the median of the maximum distance traveled by users at the national, state, and county levels. Through the Data for Good [5, 6] effort, Facebook makes available the fraction of users who stay put in a 60x60 meter tile.

By using indicators such as these, we can derive a *mobility change measure* by comparing mobility as measured by an indicator for a specific point in time and place with a *baseline* representing normal mobility as measured by the same indicator (see Table 1).

To preserve the privacy of users, many companies like Google and Apple have not made available *mobility indicators*, but instead make available their *mobility change measures*, which are based on some measured *mobility indicator*. Google [9] makes available the percent change in minutes spent at various POI groups like parks, residential, workplace, retail, and public transportation each day compared to a baseline. Similar mobility change measures based on POI visits are produced by Foursquare [11], Safegraph [7], and Unacast [14]. Apple [10] makes available the percent change in volume of direction requests. Unicast also makes available the absolute change in distance traveled compared to a baseline and the absolute change in unique human encounters per sq km. Safegraph [8] makes available their own *mobility change measure*, a shelter in place index which measures the change in percent in the population that stays at home each day compared to a baseline. Descartes Labs Inc. [4] makes available the percent change in max distance traveled relative to a baseline. In all of the above examples, the choice of baseline varies. In most cases, these baselines are static and usually look at the average mobility measured by the indicator of choice over a short period representing normal mobility (usually January or February 2020).

Quantifying the spatial and temporal change in mobility has been critical for evaluating the effectiveness of NPIs [15–17], explaining behavior related to mobility patterns [18], supporting contact tracing efforts [19], and developing realistic models that predict trajectories of the disease [20]. However, due to the challenges associated with analyzing big data with both spatial and temporal dimensions, *mobility change measures* are typically either 1) mapped to show the increase or decrease in mobility at specific points in time [13] or 2) plotted to show the change in mobility as a time series for a specific study area [21]. In either case, studies tend to associate these changes at specific points in time with certain NPIs and furthermore attempt to determine the underlying explanations for the variation in the social distancing behaviors [17]. However, A more comprehensive understanding of mobility changes is needed.

Therefore, the objectives of this study are to (1) develop a novel *mobility change measure* and (2) identify and describe common temporal and spatial trends that are observed across all US counties over the period of a year during the COVID-19 pandemic. We aim to explore three related hypotheses, as follows:

- Our first hypothesis is that mobility behavior during the pandemic varies spatially and temporally, but we can quantify general mobility trends across counties. To evaluate this

**Table 1. Publicly available mobility indicators and mobility change measures.**

| Name | Mobility Indicators | Geography | US Spatial Granularity | Mobility Change Measure | Baseline |
|---|---|---|---|---|---|
| SafeGraph [7, 8] | candidate device count, origin CBG and POI destination, completely home device count, home dwell time, non-home dwell time, distance traveled from home | US | census block group | shelter in place index, relative foot traffic index | average time spent at home/ foot traffic from Feb. 20–27, 2020 |
| Google [9] | N/A | global | county | relative time spent at various POI groups compared to baseline | median value for the corresponding day of the week, during the 5 week period Jan.3-Feb.6, 2020 |
| Apple [10] | N/A | global | county, city | relative number of direction requests compared to baseline | volume of requests on Jan. 13, 2020 |
| Foursquare [11, 12] | N/A | US | national and select states | relative number of visits to different POIs compared to baseline | average number of visits from Feb. 13–19, 2020 |
| Descartes Labs Inc [4, 13] | number of samples, median of the max distance | US | national, state, and county level | relative median max distance compared to baseline | average median max distance from Feb. 17-Mar. 7, 2020 |
| Facebook [5, 6] | fraction of users that stay put in a region | global | national, state, county, and city level | relative number of trips to other 60m tiles compared to baseline | average from Feb. 2-Mar. 29, 2020 |
| Unacast [14] | N/A | US | county level | relative distance traveled compared to baseline, relative number of visits to non essential retail and essential services compared to baseline, relative number of unique human encounters relative to baseline | average from Feb. 2-Mar. 29, 2020 |

hypothesis, we use principal component analysis (PCA) approach based on truncated Singular Value Decomposition (SVD) to decompose the mobility time series of all counties into latent components of mobility behavior.

- Our second hypothesis is that geographically close counties have similar mobility change trends. By representing each county as a linear combination of latent features of human mobility, we can map these features into geographic space and measure their spatial autocorrelation.

- Our third hypothesis is that the strength of mobility components correlate with other population characteristics such as population density, income, and political leaning. To evaluate this hypothesis, we test for a significant correlation between mobility components and these explanatory variables for the same county.

By exploring these hypotheses, we aim to uncover hidden spatial and temporal patterns and provide a concise summary of human mobility behavior.

## 2 Methods

In this paper, we analyze mobility change in the US using high-resolution foot traffic data (Fig 1). We first propose the *Delta Time Spent in Public Places* (Δ*TSPP)*, which measures changes in mobility for each US county from 2019 to 2020 (Section 2.2). In this study, any geographical region that has a FIPs code including counties, cities, and boroughs is designated as a county. Because of the high dimensionality of the data, we next use Principal Component Analysis (PCA) to reduce the data into three latent components, where each component is explained as a time series representing the change in mobility in US counties (Section 2.3).

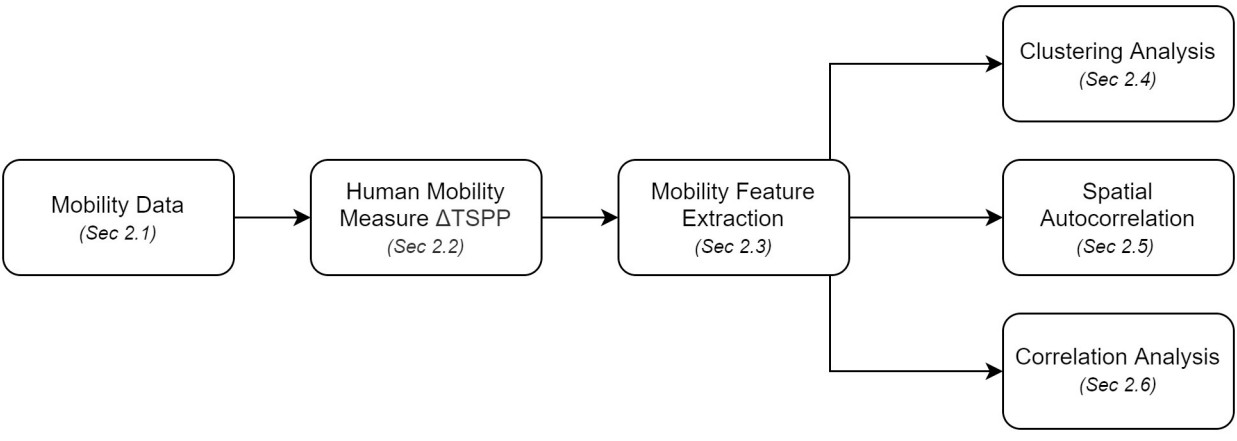

**Fig 1. Overview of methodology.**

Third, we use clustering analysis to find which counties have similar weighted combinations of the three components (Section 2.4). Fourth, we investigate local and global spatial autocorrelation for each component (Section 2.5). And finally, we conduct correlation analysis to investigate how various population characteristics and behavior correlate with mobility patterns (Section 2.6).

## 2.1 Mobility data source

The data used in this study was obtained "from SafeGraph, a data company that aggregates anonymized location data from numerous applications in order to provide insights about physical places, via the Placekey Community. To enhance privacy, SafeGraph excludes census block group information if fewer than five devices visited an establishment in a month from a given census block group" [3, 22].

As part of SafeGraph's Data for Academics [23], SafeGraph offers a Social Distancing Metrics Dataset [8] (archived on April 16, 2020). This dataset is available at no cost for academic researchers for non-commercial work. In general, SafeGraph obtains precise device location information from third-party data partners such as mobile application developers. This information is collected through APIs where app developers provide information about their users [24]. The device's home census block group (CBG) is determined based on the nighttime location of devices over six weeks so that Social Distancing Metrics including the CBG's device count, the complete at home device count, the median distance traveled from home, the median home dwell time, median non-home dwell-time and more can be calculated. It should be noted that SafeGraph's Social Distancing Metrics are available from January 1, 2019, through to April 16, 2021, and are no longer being updated. For this study, we use data from SafeGraph's Social Distancing Metrics from January 1, 2019, to December, 31st 2020.

## 2.2 Human mobility measures

As an indicator to assess changes in human mobility in the US, we selected SafeGraph's measure of `median_non_home_dwell_time` from the Social Distancing Metrics dataset, which is defined as the median time (minutes each day) that all devices in a census block group (CBG) spend visiting public points of interest (POIs) located outside the boundaries of their home geohash (using a $153m \times 153m$ hash buckets). This includes minutes spent at

public POIs such as grocery stores, restaurants, bars, and movie theaters. We note that this measure only includes public POIs that are captured among the 6.5 million POIs in the Safe-Graph Core Places database [7]. We aggregate this measure to the county level by averaging the `median_non_home_dwell_time` for each CBG in each county to give us an `average_median_non_home_dwell_time` for each county.

Our goal is to compare the `average_median_non_home_dwell_time` for each county for each day in 2019 and the corresponding day in 2020. Weekly patterns strongly affect mobility in the United States, where in general, lower mobility is observed on the weekends, especially on Sundays. Since corresponding days in 2019 and 2020 may not be corresponding days of the week, we use the seven-day rolling average of the `average_median_non_home_dwell_time` for both 2019 and 2020. That is, we calculate the mean `average_median_non_home_dwell_time` for that day and the three days before and after that day. We formally define this as Time Spent at Public Places (TSPP) calculated for both 2019 and 2020 as follows:

**Definition 1 (Time Spent at Public Places (TSPP))** *Let $\mathcal{R}$ be a region, and let $\mathcal{D}^{\mathcal{R}} = [d_1^{\mathcal{R}}, ..., d_N^{\mathcal{R}}]$ denote a time series of daily* `median_non_home_dwell_time` *for that region, where N is the number of days of interest. We define our Time Spent at Public Places measure at the i-th day as*:

$$TSPP(\mathcal{D}^{\mathcal{R}})[i] = \frac{\sum_{j=-3}^{3} d_{i+j}^{\mathcal{R}}}{7},$$

where *i* is an index of $D^{\mathcal{R}}$ and $4 \leq i \leq N - 3$. For example, for a 365 day time series, $TSPP(\mathcal{D}^{\mathcal{R}})[i]$ is defined from Day 4 to Day 362, since for the first and last three days, there are not enough day before and after, respectively, to compute the centered weekly average. The denominator denotes the size of a sliding window, i.e., 7 days, used for calculating the mean.

Next, we look at the difference between the Time Spent at Public Places (TSPP) measure calculated for the *i-th* day in the time series for each county in 2019 and the *i-th* day in the time series for the same county in 2020. We define this as Change in Time Spent at Public Places measure or ΔTSPP as follows:

**Definition 2 (Time Spent at Public Places (ΔTSPP))** *Let $\mathcal{R}$ be a region and let $\mathcal{D}_y^{\mathcal{R}}$ the time series of 365 daily* `median_non_home_dwell_time` *values in $\mathcal{R}$ for all days in year y. We define the daily change in TSPP as*:

$$\Delta TSPP(\mathcal{R}, y1, y2) = TSPP(\mathcal{D}_{y1}^{\mathcal{R}}) - TSPP(\mathcal{D}_{y2}^{\mathcal{R}}),$$

where *y*1 and *y*2 are a target year and a reference year to compare, respectively. For short, ΔTSPP is referred to $\Delta TSPP(\mathcal{R}, 2020, 2019)$ in this paper, where $\mathcal{R}$ is all counties of the US.

We consider an increase in ΔTSPP, where TSPP is higher in 2020 than in 2019, as a proxy for increased mobility, thus increasing the risk of exposure. Although individual counties can provide the spatial and temporal heterogeneity in the post-pandemic mobility behavior, there are thousands of counties in the US, each with a unique mobility trend. Thus, we seek an approach that can identify different mobility trends found commonly across all 3107 counties while handling both the dimensionality and variance of the data.

## 2.3 Mobility feature extraction: Principal Component Analysis

Principal Component Analysis (PCA) [25] is a commonly used technique to reduce the dimensionality, yet maintain the variation, present in large multivariate data and is a generalization of eigendecomposition for non-square and non-maximum rank matrices. We define a

data matrix $R \in \mathbb{R}^{m \times n}$ as a $m \times n$ matrix where $m = 359$ corresponds to days the number of days of the year (except for the first three and last three days due to the seven-day sliding window) and where $n = 3100$ corresponds to the number of US counties (we remove seven outlier counties—see Section 2 in S1 File). Using singular value decomposition (SVD), $R$ is factorized in the product of three matrices $R = U\Sigma V^T$ where $\Sigma$ is a diagonal matrix containing the square roots of the eigenvalues of $RR^T$, and the columns of $U$ ($V$) are the eigenvectors of $RR^T$ ($R^T R$).

To reduce the dimensionality of $R$ we truncate the SVD to obtain only the first $K$ dimensions. Thus, $\Sigma_K$ is a $K \times K$ diagonal matrix containing the $K$ largest eigenvalues, $U_K$ is a $m \times K$ matrix describing each of the $m$ days with $K$ latent features, and $V_K$ is a $K \times n$ matrix describing each county with $K$ latent features.

The idea of using SVD in this context is to decompose the time series of each county into a linear combination of $K$ archetypal time series called principal components (PCs). SVD assumes that the ΔTSPP is a linear combination of latent features. This assumption holds since the average ΔTSPP that we observe is indeed derived from the mobility change of individual people. By applying SVD to the set of ΔTSPP time-series of all counties of the US, we can find components of individual human behavior as follows.

- Reduced mobility during the entirety of March-December 2020 corresponding to counties with individuals who have the ability and obedience to stay at home for the remainder of 2020, such as people who worked remotely.

- Reduced mobility only during Summer 2020 corresponding to individuals who stop isolation after the first wave of infections in the US, either due to having to go back to work or due to growing weary of mobility restrictions.

- No reduced mobility, corresponding to individuals who cannot stay at home such as health professionals or individuals who are not willing to follow stay-at-home directions.

In addition to finding these three latent PCs (see Section 3.3 in S1 File), SVD further allows us to describe each county as a linear combination of these components, which can be interpreted as corresponding mobility behavior. In the case that some counties are not well explained by any of the latent components, we calculate the coefficient of determination for each county.

## 2.4 Clustering analysis

Due to a large number of counties, it is difficult to determine which counties exhibit similar mobility trends. Therefore, we cluster counties into groups of counties that exhibit similar latent features of change of exposure. We first plot each county into the PCA space where each point represents a county, and each axis represents the weight of each PC in explaining the county's ΔTSPP, normalized from 0 to 1. For clustering, we compared the K-means algorithm [26] against other clustering algorithms and determined K-means to be the most appropriate (see Section 3.4 in S1 File). The K-means algorithm partitions $n$ observations into $k$ clusters by randomly initializing $k$ points (means or cluster centroids) and assigning each observation to their closest point. The coordinate point is updated iteratively to reflect the mean center of observations that belong to it. This approach requires the number of $k$ points to begin with. We choose $k = 3$ so that we can better visualize the counties that have similar weighted combinations of PCs (see Section 3.4 in S1 File for more details).

## 2.5 Spatial autocorrelation analysis

To test the impact of proximity in mobility change, we measure the spatial autocorrelation of counties and their corresponding weights for each PC using both Global Moran's I [27] and Anselin's Local *Moran's I* [28]. The concept of spatial autocorrelation is based on Tobler's First Law of Geography which states that things that are closer to each other are more similar than things that are far apart [29]. *Moran's I* calculates the degree to which features in a dataset are positively spatially autocorrelated (neighboring features are alike), negatively spatially autocorrelated (neighboring features are not alike), or not spatially autocorrelated (attributes of features are independent of location).

First, we compute a matrix of spatial weights to define each counties' neighbors mathematically. We used Queens-case, meaning counties are considered to be neighboring if their border shares at least one common vertex. After building this matrix, we discovered that the only neighboring county to Fairfax City is Fairfax County, Virginia, which was identified as an outlier in the PCA space and removed earlier in this analysis. To resolve this issue, Fairfax City was also removed. Next, we calculate the fraction of the total variation that is attributed to counties that are close together across the entire study area to give us a measure of global spatial autocorrelation (Moran's I) and then decompose the measure for each feature to give us local spatial autocorrelation (Anselin's Local Moran's I).

## 2.6 Correlation analysis

We have covered both the spatial and temporal variation of mobility trends across all of the counties in the US in response to the COVID-19 pandemic. Next, we aim to identify some population variables that may explain the variation. Thus, we use the Pearson's R coefficient to calculate the correlation between the weight of each PC and a variety of explanatory variables, including income, political leaning, employment, percent age over 65, and COVID-19 cases and deaths for each county. Pearson's R is a statistical measure of linear association that returns a value between -1 and 1 that defines how strong the correlation is, where the further away that value is from 0, the stronger the correlation. We test for significance using a p-value. Since we hypothesize that there is a linear relationship between the strength of the PCs in explaining county mobility and different county variables, we did not investigate non-linear relationships.

## 3 Results

### 3.1 General mobility trends

We calculate the TSPP for each of the 3107 counties to produce a time series representing mobility in each county in 2019 and 2020. This can be aggregated to the US. Fig 2 shows the Time Spent in Public Places $TSPP(\mathcal{D}^{\mathcal{R}})$ for the region $\mathcal{R}$ corresponding to all counties aggregated to the United States level, excluding Alaska, Hawaii, and US territories, and for the sequence of days $\mathcal{D}^{\mathcal{R}}$ ranging from Jan to Dec. for 2019 and 2020. The boxplots for the 2019 and 2020 TSPP can be found in the Section 3.1 in S1 File.

We observe anomalously high mobility in January and February 2020. This is likely a combined effect due to higher-than-average temperatures, 50% less snow depth, and panic buying behaviors (see S1 File). Starting March 2020, we observe a rapid drop in mobility due to the COVID-19 pandemic. Interestingly, we also observe that these drops swing back to normal by June 2020 and even exceeds 2019 mobility overall.

Next, we look at the Change in Measure of Public Exposure. Fig 3 shows the ΔTSPP measure for the US and for three counties. We can visually observe radically different mobility

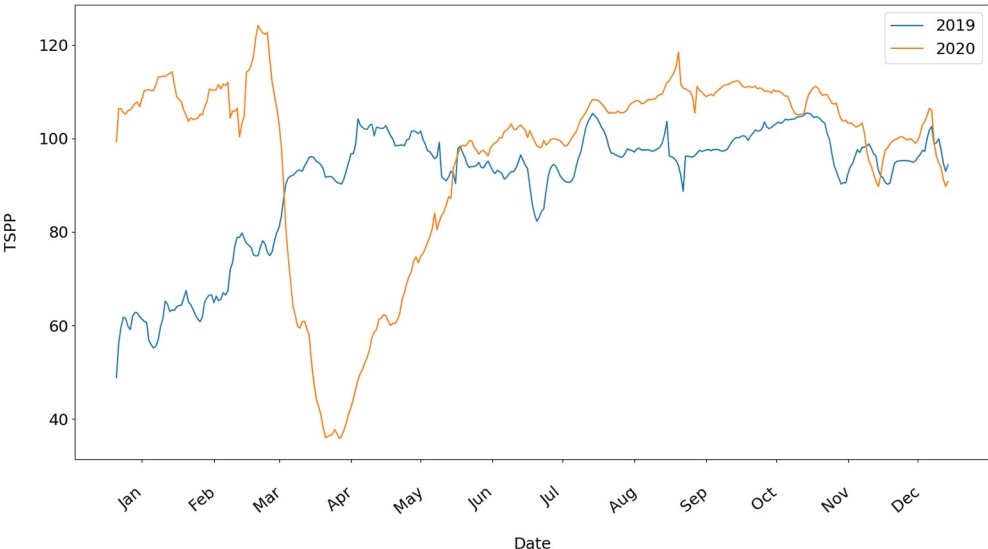

**Fig 2. TSPP calculated for the United States in 2019 and 2020.**

behavior among these counties. Arlington County, VA, exhibits a large drop in mobility in March 2020 than other counties. We also observe that this reduction in mobility persists throughout the year 2020. In contrast, the mobility of Cambria County, PA, exhibits a less extreme drop in mobility and quickly returns to and exceeds normal mobility after June 2020, where ΔTSPP is greater than or equal to 0. Tulare County, California, exhibits a much less extreme drop in mobility, but in general, maintains this reduction of mobility.

## 3.2 Principal components of ΔTSPP

**3.2.1 Qualitative interpretation.** We find that $K = 3$ principal components explain 59% of the variation in all of the included time series where PC1 explains 35.6%, PC2 explains

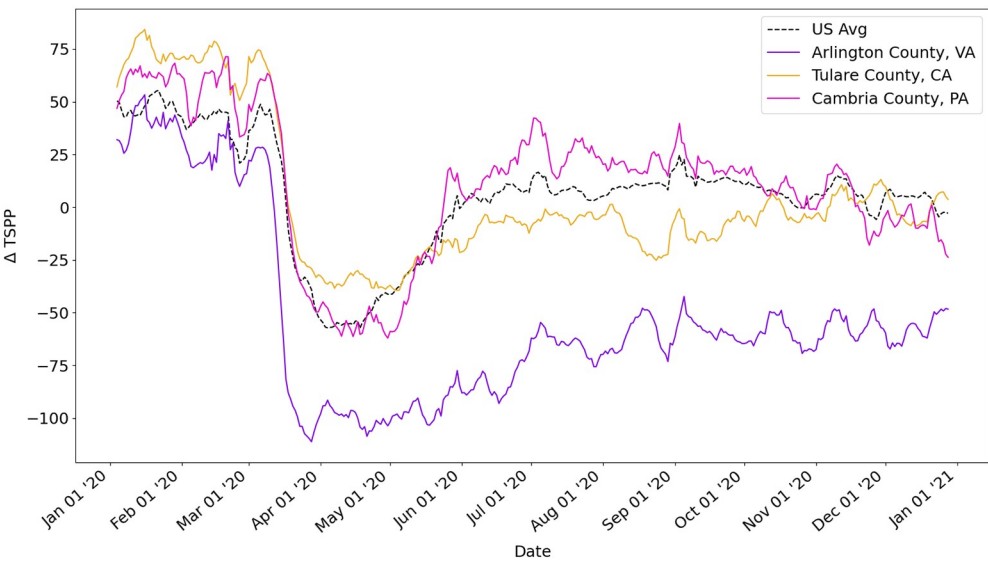

**Fig 3. ΔTSPP calculated for the US, Arlington County (VA), Tulare County (CA), and Cambria County (PA).**

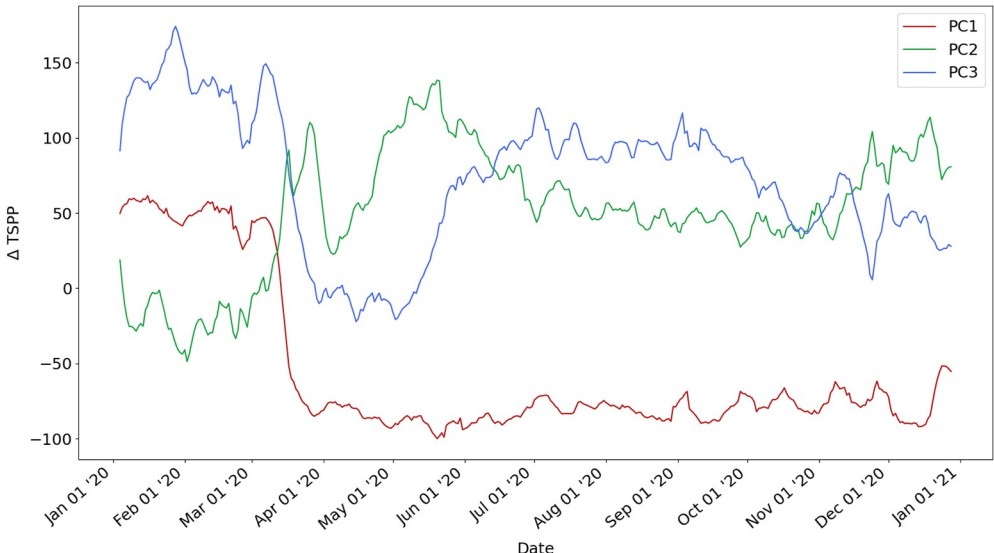

**Fig 4. Three latent features describing mobility in the US mapped back into the temporal space.**

15.3%, and PC3 explains 8.8% of the variance. Thus, using Matrix $V_K$, each county in the U.S. is described as a linear combination of three PCs, with a loss of 1–59% = 41% of explained variance (see Section 3.3 in S1 File). This result shows that by describing each county by only three archetypal behaviors, we can explain more than half of the variance across the 359 dimensions that each county is described by in the full space.

Towards explainable machine learning, Fig 4 shows the three latent principal components describing each county in the U.S. mapped back into the full 359-day space.

- The mobility trend captured by PC1 (in red) begins with slightly higher mobility in January and February 2020 compared with 2019. In March 2020, there is a sharp deviation from the mobility observed in March 2019 as mobility declines in response to the pandemic. For the remainder of 2020, mobility remains consistently lower than mobility in 2019. We explain PC1 by individuals who reduced their mobility in March 2020 and then maintained this stay-at-home and social distancing behavior for the rest of the year. Counties that are well explained by PC1 may be composed of individuals who are able to work from home in April and continued working from home throughout the year.

- The mobility trend captured by PC2 (in green) begins with a slightly lower mobility in January and February 2020 than in comparison with 2019. In the spring, mobility steadily increases until April 2020, when mobility declines slightly in response to the COVID-19 pandemic. For the remainder of 2020, mobility remains higher than mobility in 2019. The only time mobility is lower than in comparison to 2019 is before the pandemic. Counties that are well explained by PC2 may be composed of individuals who *can't* or *won't* comply with stay-at-home orders (such as health care workers, essential workers, and other individuals).

- Finally, the mobility trend captured by PC3 (in blue) begins with more mobility in January and February 2020 than in 2019. In response to the pandemic, there is a sharp drop in mobility in March 2020 until it returned to normal mobility in June 2019. Mobility then increases in late summer and remains higher for the remainder of the year than in 2019. We explain PC3 by individuals who have reduced mobility directly after the pandemic (March-June

2020) but then return to normal mobility. Counties that are well explained by PC3 may be composed of individuals who were unable to work due to a shutdown in March 2020, but returned to work in June 2020. Counties that are well explained by PC3 may also be composed of individuals who were fearful during the onset of the pandemic but experienced pandemic fatigue and became less compliant with stay-at-home orders as the pandemic continued.

Abstractly speaking, we can interpret these three latent components as "Long term mobility reduction" (PC1), "No mobility reduction" (PC2), and "Short term mobility reduction" (PC3).

In the Singular Value Decomposition, Matrix $V$ provides us with the $K = 3$ weights of these latent features for each county. In the following, Section 3.2.1 provides a qualitative spatial analysis of these three principal components to understand which parts of the United States exhibit strong weights for each of these latent features. We provide quantitative analysis to show that the weights of these latent features are strongly spatially auto-correlated with a number of significant spatial clusters.

**3.2.2 Spatial analysis.** We provide a spatial analysis of the principal components of change of exposure across all counties in the United States. Section 3.4 shows the spatial distribution of each principal components, Section 3.2.2 analyses clusters of counties having similar principal components, and Section 3.5 explores which counties are well-modeled by these components and which counties still exhibit large unexplained variance using three principal components only.

Matrix $V$ describes each county as a linear combination of the three components. For example, Arlington County's ΔTSPP (See Fig 3) is described as 98% PC1, 20% PC2, and 21% PC3, thus having a dominant first component. In contrast, Tulare County's ΔTSPP (See Fig 3) is described as 51% PC1, 16% PC2, and 33% PC3, thus having a stronger weight on PC3 than Fairfax.

Fig 5A to 5C maps the strength of each PC in explaining the mobility trend of all 3100 counties. Based on visual analysis of the results, we find that the counties on the east and west coast have a higher weight in PC1, counties in the south-west and south-east have a higher weight in PC2, and counties in the mid-west have a higher weight in PC3. Stacking these three figures creates a Red-Green-Blue (RGB) composite map to show the linear combination of the components for each county where red is PC1, green is PC2, and blue is PC3 (see Section 3.5 in S1 File).

Since the three principal components only explain 59% of the variation among the 358-dimensional representation of counties as a sequence of daily changes in mobility, an important and open question is to ask which counties of the U.S. are explained well by these components and which ones are not. That is, which counties may be better explained by the remaining 355 components that we truncated to reduce the dimensionality. Fig 6 shows the explained and unexplained variance using the coefficient of determination. The counties with positive values (green) are well explained by the three PCs. The counties with negative values (red) are not well explained by the three PCs and would be better explained by taking the simple average of the counties ΔTSPP.

## 3.3 Clustering of latent features of ΔTSPP

Fig 7 depicts the resulting feature vectors for all counties in the $K = 3$ dimensional latent feature space from two angles (Fig 7A and 7B) for easier interpretation. The colors in Fig 7 represent the result of the K-means clustering analysis. We map the results of K-means analysis to see the spatial distributions of the counties related to each cluster (Fig 8).

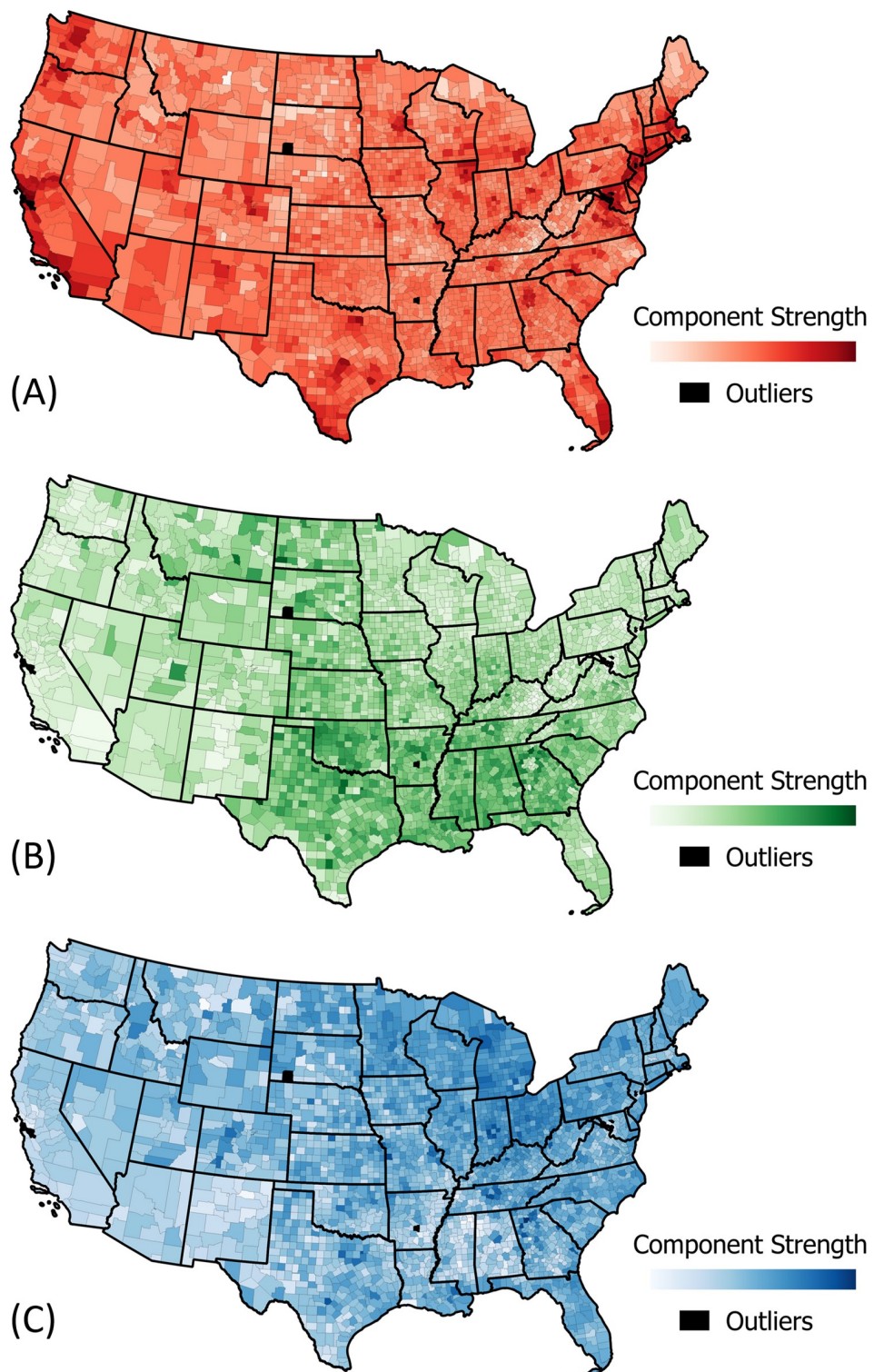

**Fig 5. Spatial distribution of the three principal components of change of public exposure.** (A) Principal Component 1: Reduced ΔTSPP March-December (B) Principal Component 2: Increased ΔTSPP March-December (C) Principal Component 3: Reduced ΔTSPP March-June. Maps produced in QGIS [30] using SafeGraph [3] derived data, shapefiles from data.gov [31].

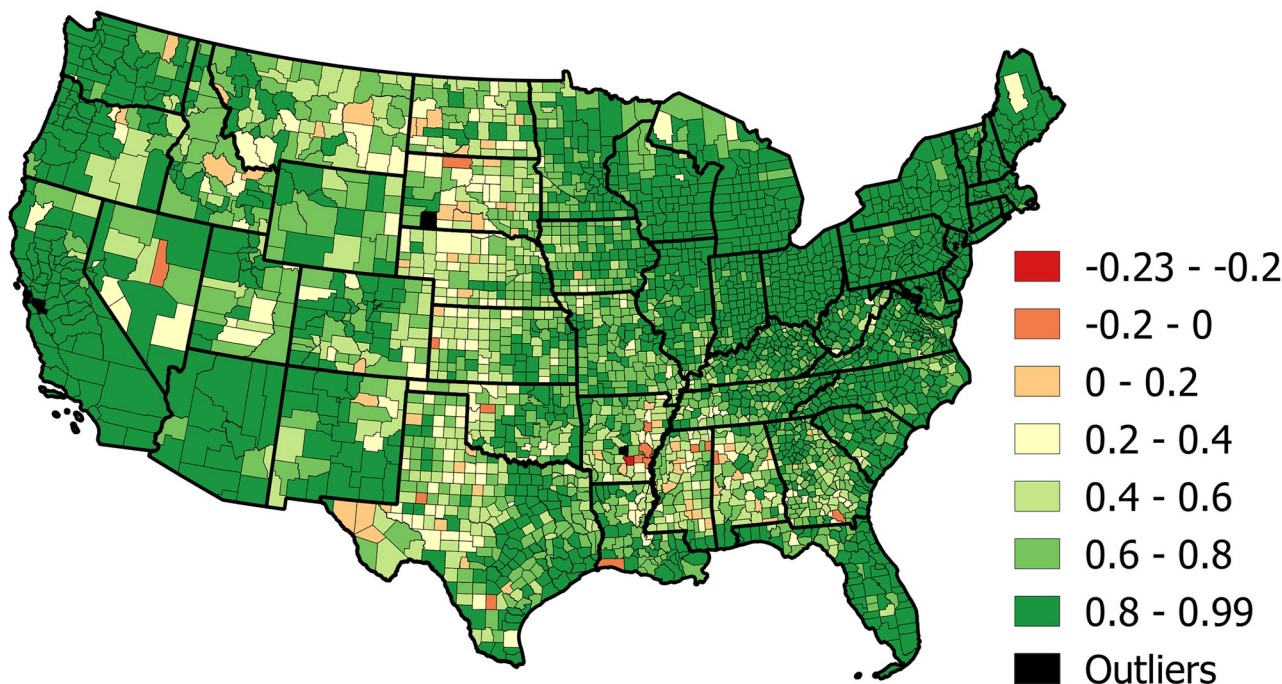

**Fig 6. Spatial distribution of the explained variance ($R^2$) for all counties across the US using three principal components.** Map produced in QGIS [30] using SafeGraph [3] derived data, shapefile from data.gov [31].

We can see that counties in the southwest and southwest, excluding Arizona, New Mexico, Virginia, and West Virginia, belong primarily to Cluster 1 (in green) and thus have similar weighted combinations of PCs. Counties along the west and northeastern coast, as well as the Florida coast and southern Texas, belong primarily to Cluster 2 (in pink). Counties that belong to the Rocky Mountain region of the US as well as Maine, Vermont, much of New York, West

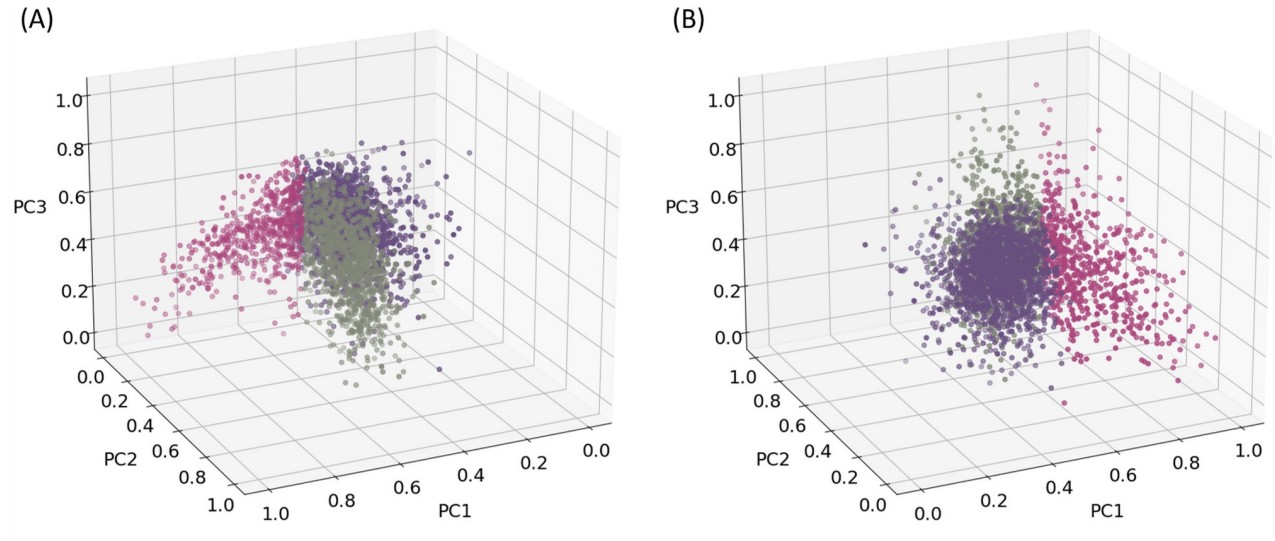

**Fig 7. Counties plotted in PCA space.** (A) Angle 1. (B) Angle 2.

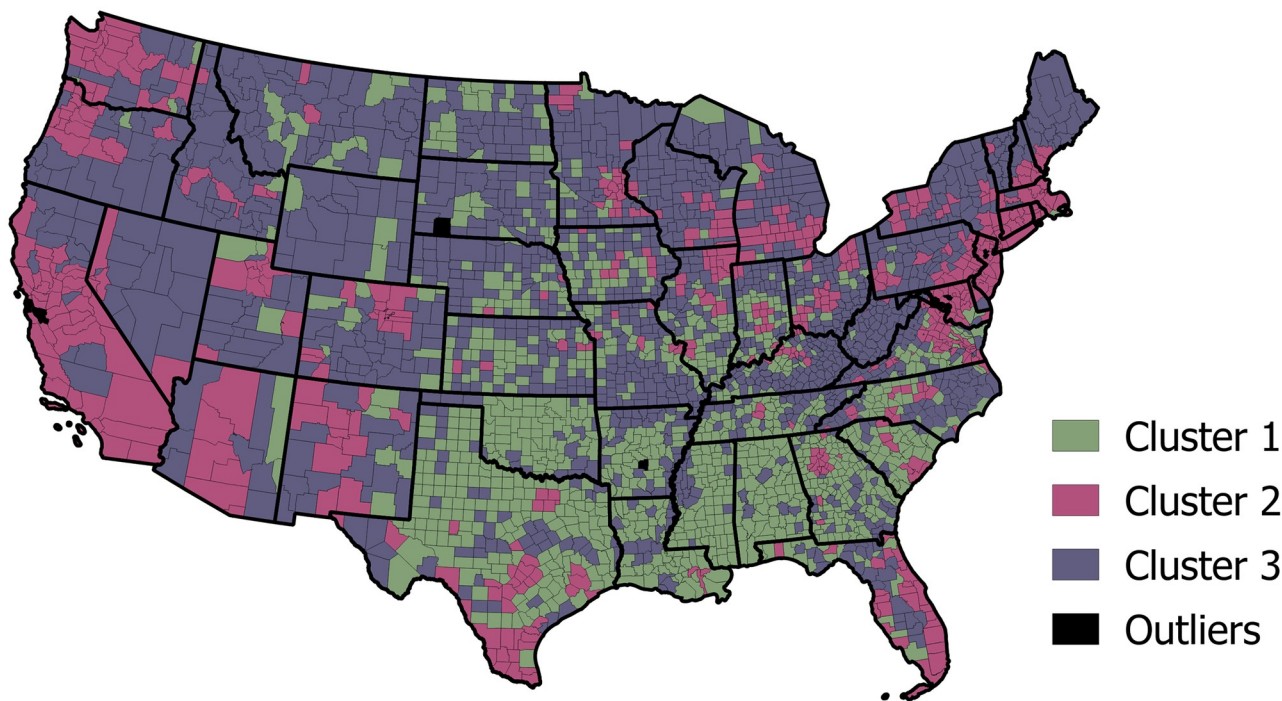

**Fig 8. K-means analysis mapped geographically.** Map produced in QGIS [30] using SafeGraph [3] derived data, shapefile from data.gov [31].

Virginia, Minnesota, and Wisconsin belong primarily to Cluster 3 (in purple). Counties in many states in the midwest and southeast are a mix of belonging to different clusters.

## 3.4 Spatial autocorrelation of mobility behavior between counties

The results of the Global Moran's I analysis for each of the three latent features are presented in Table 2. We found a strong positive spatial autocorrelation for all three features. The spatial autocorrelation of PC1 and PC2 at 0.556 and 0.544, respectively, is higher than the spatial autocorrelation of PC3 at 0.489. For all three components, the positive spatial autocorrelation is highly significant at p-values $\ll 10^{-28}$ having Z-scores of 46 and greater. As we suspected from our qualitative analysis, this result confirms that the patterns of $\Delta$TSPP that we observed in Fig 5 are indeed strongly positively spatially autocorrelated.

Next, we calculate Anselin's Local Moran's I [28] to help visualize clusters of counties with similar neighbors and outlier counties with dissimilar neighbors (Fig 9). The results of Anselin's Local Morain's I for PC1 are presented in Fig 9A. We can identify clusters of counties with high weights corresponding to PC1 that have neighbors with high weights. We refer to these patterns as High-High (HH) clusters that are positively spatially autocorrelated. In addition, we can identify the counties with low weights corresponding to PC1 that have neighbors

**Table 2. Global Moran's I measure of spatial autocorrelation for each principal component.**

| Component | Moran's I | Z-score |
|:---:|:---:|:---:|
| PC1 | 0.556 | 52.4 |
| PC2 | 0.544 | 51.3 |
| PC3 | 0.489 | 46.1 |

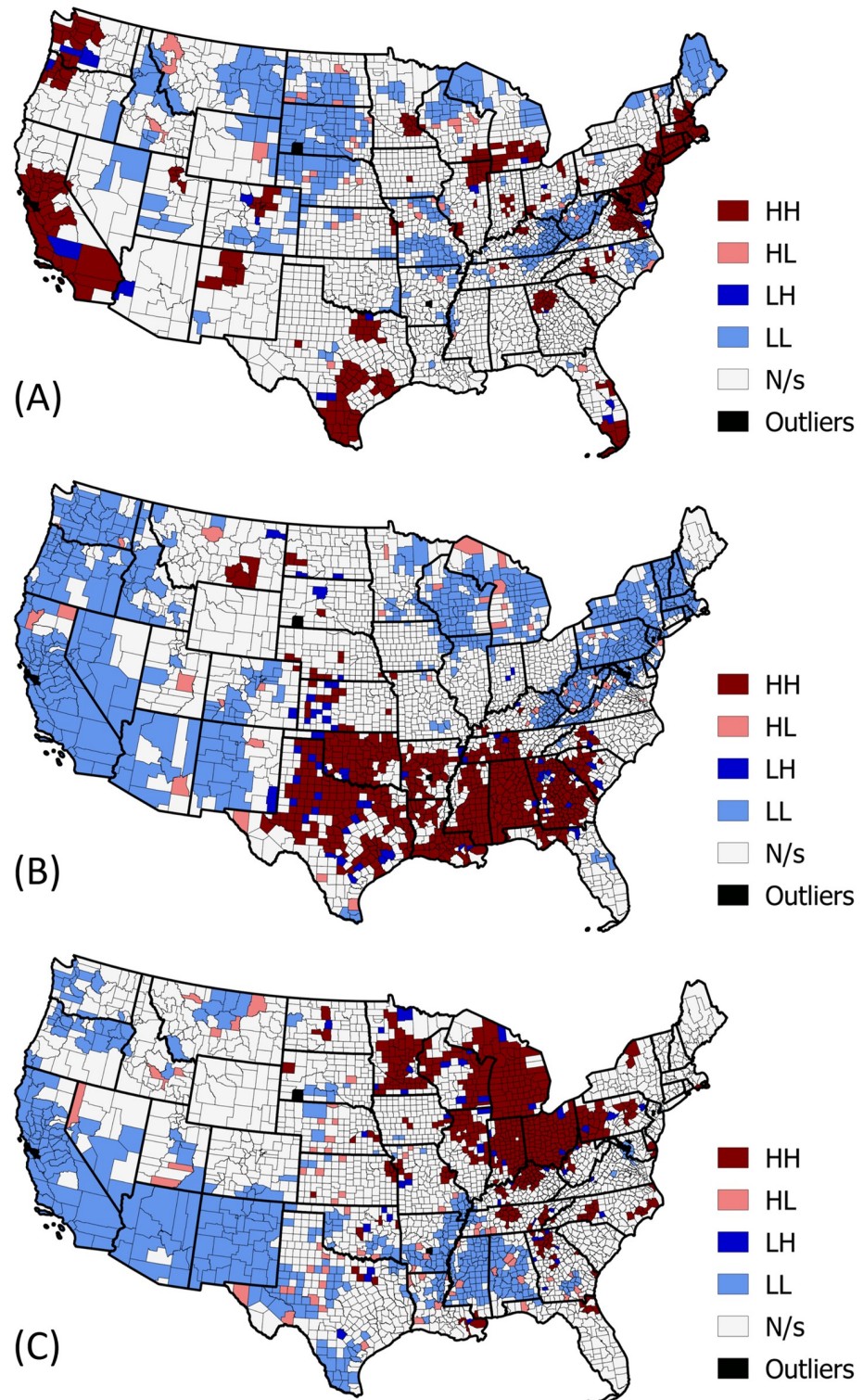

**Fig 9. Anselin's Local Moran's I (LISA) results.** (A) LISA for PC1. (B) LISA for PC2. (C) LISA for PC3. Maps produced in QGIS [30] using SafeGraph [3] derived data, shapefiles from data.gov [31].

with low weights. We refer to these patterns as Low-Low(LL) clusters that are positively spatially autocorrelated. Anselin's Local Moran's I also uncover outliers, where we find counties with high or low weights corresponding to PC1 that have oppositely weighted neighbors. We refer to these patterns as Low-High(LH) outliers and High-Low (HL) outliers that are negatively spatially autocorrelated. Counties with a p-value of greater than.05 are considered insignificant. We present the results of Anselin's Local Moran's I for PC2 and PC3 in Fig 9B and 9C.

## 3.5 Explaining variation in mobility patterns

We examine the correlation between the weight of each PC and other variables for each county. The results are presented in Table 3. PC1 captures the mobility trends in US counties that maintain decreased mobility from the onset of the pandemic and beyond. We find that there is a strong positive correlation between counties with a higher income (median household income and per capita income) and a higher weight corresponding to PC1. This has been supported in the literature in other studies that find that higher-income counties and states are able to follow social distancing guidelines better and stay-at-home orders [32].

We find a strong positive correlation between counties that are democratic leaning and have a higher weight corresponding to PC1. In contrast, we find a strong negative correlation between counties that are republican leaning in the 2020 and 2016 election and have a higher weight corresponding to PC1. This has been supported in the literature where it has been found that counties and states that are democratic leaning better follow social distancing guidelines and stay-at-home orders [33]. Interestingly, we find that the strength of the correlations between PC1 weight and political leaning decrease as we use political data from 2012, 2008, 2004, and 2000.

**Table 3. Correlation analysis results showing Pearson's correlation and respective p-values between each principal component (PC1-PC3) and other population characteristics.**

| Correlation Variable | PC1 | PC1 P-value | PC2 | PC2 P-value | PC3 | PC3 P-value |
|---|---|---|---|---|---|---|
| Median Household Income | 0.63 | 0.00e+00 | -0.11 | 1.84e-09 | 0.22 | 2.39e-36 |
| Per Capita Income | 0.55 | 1.02e-244 | -0.15 | 2.02e-16 | 0.19 | 5.13e-26 |
| 2020 Rep Vote Percent | -0.45 | 8.12e-152 | 0.38 | 2.90e-104 | -0.02 | 3.34e-01 |
| 2020 Dem Vote Percent | 0.44 | 3.61e-150 | -0.37 | 4.56e-99 | -3.23e-02 | 5.14e-01 |
| 2016 Rep Vote Percent | -0.42 | 5.79e-133 | 0.39 | 7.45e-112 | 0.02 | 1.72e-01 |
| 2016 Dem Vote Percent | 0.41 | 5.40e-125 | -0.33 | 1.93e-81 | -0.03 | 7.25e-02 |
| ACS 2019 Pop Est | 0.38 | 4.50e-106 | -0.18 | 3.32e-25 | -0.02 | 2.32e-01 |
| Percent Pop Over 65 | -0.34 | 1.23e-84 | -0.025 | 0.16 | -0.078 | 1.50e-05 |
| 2012 Dem Vote Percent | 0.28 | 2.42e-56 | -0.39 | 1.11e-115 | 0.02 | 3.38e-01 |
| 2012 Rep Vote Percent | -0.28 | 4.62e-55 | 0.42 | 1.07e-130 | -0.02 | 3.55e-01 |
| 2008 Dem Vote Percent | 0.26 | 2.18e-47 | -0.44 | 2.48e-143 | 0.05 | 5.09e-03 |
| 2008 Rep Vote Percent | -0.24 | 1.03e-41 | 0.45 | 1.32e-157 | -0.05 | 4.78e-03 |
| 2000 Rep Vote Percent | -0.19 | 7.16e-27 | 0.30 | 1.99e-64 | 0.03 | 1.01e-01 |
| 2004 Dem Vote Percent | 0.19 | 1.01e-25 | -0.35 | 2.06e-89 | -1.49e-02 | 4.07e-01 |
| 2004 Rep Vote Percent | -0.18 | 3.74e-25 | 0.36 | 1.65e-97 | 0.02 | 3.41e-01 |
| 2000 Dem Vote Percent | 0.17 | 1.95e-21 | -0.23 | 4.50e-40 | -0.03 | 9.62e-02 |
| Unemployment Rate | 0.11 | 1.42e-10 | -0.31 | 1.28e-69 | -0.02 | 2.65e-01 |
| Deaths Per Thousand | 0.09 | 1.36e-07 | 0.15 | 2.64e-16 | -0.17 | 1.31e-21 |
| Cases Per Thousand | -0.05 | 1.18e-02 | 0.25 | 5.15e-44 | -0.16 | 6.41e-20 |

We also find moderate positive correlations between counties with high population and a higher weight corresponding to PC1. We find moderate negative correlations between counties with high percentage of population over 65 and a higher weight corresponding to PC1. We do not find strong correlations between counties with a higher weight corresponding to PC1 and normalized numbers of cases and deaths corresponding to COVID-19. In any case, it is difficult to properly quantify the relationship between total and normalized COVID-19 cases and deaths and PC weight based on the uncertainty inherent to the data due to inconsistent reporting.

PC2 captures the mobility trends in US counties that increase their mobility. We find that the correlations between counties with a high weight of PC2 and the variables are opposite that of PC1. Thus, there is a strong positive correlation between counties that are republican leaning in the 2020 and 2016 election with a higher weight corresponding to PC2. PC3 captures the average mobility trends in US counties. There does not appear to be nearly as strong correlations between counties with a high weight corresponding to PC3 and the variables.

## 4 Discussion and conclusions

In this study, we calculate a novel indicator of mobility change which we call the *Time Spent at Public Places (TSPP)* using Safegraph data [3]. We quantify the change of mobility by calculating the running difference of this measure between 2019 and 2020 for each county to estimate a measure of mobility change (ΔTSPP) that describes each county as a time series of mobility change of 365 days.

Confirming our first hypothesis, we find that mobility behavior during the pandemic varies spatially and temporally, but that there are three main mobility trends uncovered by our PCA analysis. PC1 captures the mobility trends of counties that drop their mobility at the onset of the pandemic and maintain reduced mobility through to the end of the year, the behavior of which we refer to as "Long term mobility reduction". PC2 captures the mobility trends of counties that increase their mobility in 2020 or, in other words "No mobility reduction". PC3 captures the mobility trends of counties that drop their mobility at the onset of the pandemic and then quickly return to normal mobility, which we call "Short term mobility reduction". PC3 can be considered the average mobility trend across all counties. Confirming our second hypothesis, we find that mobility trends are positively spatially autocorrelated, meaning that counties that are geographically close exhibit similar mobility trends. Finally, we partly confirmed our third hypothesis is that we find some correlations between the variation in mobility trends and underlying population behavior and characteristics.

While we obtained interesting results, there are certain limitations of the data relevant to this study including sparse documentation of data collection, data completeness, bias, geographic coverage, and open-access. First, although SafeGraph has gone to unprecedented lengths to make the data public, perhaps unsurprisingly as a corporate data provider, their methods and sources for collecting device data and POI data are sparse. Detailed methods, sources of data, and truth datasets are not available and thus cannot be independently evaluated [34].

Data completeness is also difficult to assess. Our mobility indicator is based on the `median_non_home_dwell_time` which measures the median time that devices in the same CBG spend at public POIs that are included in SafeGraph's Core Places database. SafeGraph represents the location of over 6300 distinct brands as POIs and this number changes over time as new brands are added. These are chains of commercial POIs that include all major brands in the United States (McDonald's, AMC, Macy's, Chevrolet, Whole Foods Market). Of the brands that SafeGraph includes, they capture close to 100% of the brands' locations [35].

About 80% of SafeGraph POIs have no brand associated as they are single commercial locations (local restaurants, museums). It is not possible to assess the actual completeness of SafeGraph's POIs; specifically, the total number of all POIs in the US versus the total number of POIs represented in the Safegraph Core Places Dataset.

SafeGraph's Social Distancing Metrics dataset is based on device users that make up approximately 10% of the United States population, which is significantly larger than typical household mobility surveys. Although there are concerns that the sample is not a perfect representative of the population, SafeGraph reports that their sample correlates very highly with true census populations [36]. SafeGraph finds a Pearson's R Correlation Coefficient of 0.966 and a Sum Absolute Bias of 24.77 between the real county population and the number of devices in the county counted by SafeGraph. SafeGraph finds little to no race-level sampling bias, educational attainment-level sampling bias, and household income-level bias with Pearson's R Correlation Coefficients of 1, 0.999, and 0.997 and Sum Absolute Biases of 3.70, 3.43, and 1.75, respectively. The code to run independent sampling bias analysis on SafeGraph data is provided by SafeGraph [37].

We found that the geographic coverage of the data was complete at the county level with 100% coverage. There were no counties that were excluded from the dataset as a result of few POIs or lower device counts (see the Section 3.6 in S1 File). As part of SafeGraph's Data for Academics, academic researchers have no-cost access to SafeGraph data for non-commercial work. The reliance on commercial data means that there are limited safeguards to the data, and changes to data and data access may occur beyond our control. For example, SafeGraph recently stopped updating the Social Distancing Metrics dataset. However, SafeGraph provided ample notice, and the archived data is still available for academic researchers, ensuring the reproducibility of this study. Furthermore, we have made available ΔTSPP on this project's GitHub Repository (see the Section 1 in S1 File).

Our results are only applicable to the United States. Application of the same methodology to other countries is yet to be conducted. Finally, our PCA components capture ≈59% of the movement patterns, and the rest 41% is unexplained with our approach. Additional work is needed to cover a better percentage of variation without significantly increasing the PCA space. Future work will also focus on exploring the correlation between the PCs and additional variables, including commute, weather, and policy guidelines. This study provides a more comprehensive and data-driven approach to examining how human mobility has changed in response to the pandemic.

## Supporting information

**S1 File.**
(PDF)

## Acknowledgments

We thank the Editor and the two reviewers for their valuable feedback.

## Author Contributions

**Conceptualization:** Justin Elarde, Joon-Seok Kim, Hamdi Kavak, Andreas Züfle, Taylor Anderson.

**Data curation:** Justin Elarde.

**Formal analysis:** Justin Elarde, Joon-Seok Kim, Hamdi Kavak, Andreas Züfle, Taylor Anderson.

**Funding acquisition:** Hamdi Kavak, Andreas Züfle, Taylor Anderson.

**Investigation:** Justin Elarde, Joon-Seok Kim, Hamdi Kavak, Andreas Züfle, Taylor Anderson.

**Methodology:** Justin Elarde, Joon-Seok Kim, Hamdi Kavak, Andreas Züfle, Taylor Anderson.

**Project administration:** Taylor Anderson.

**Software:** Justin Elarde.

**Supervision:** Hamdi Kavak, Andreas Züfle, Taylor Anderson.

**Validation:** Justin Elarde, Joon-Seok Kim, Hamdi Kavak, Andreas Züfle, Taylor Anderson.

**Visualization:** Justin Elarde.

**Writing – original draft:** Justin Elarde, Joon-Seok Kim, Hamdi Kavak, Andreas Züfle, Taylor Anderson.

**Writing – review & editing:** Justin Elarde, Joon-Seok Kim, Hamdi Kavak, Andreas Züfle, Taylor Anderson.

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
