## [Decision Letter · Decision Letter 0]

12 Apr 2021

PONE-D-21-10041

Change of human mobility during COVID-19: A United States case study

PLOS ONE

Dear Dr. Anderson,

Thank you for submitting your manuscript to PLOS ONE. After careful consideration, we feel that it has merit but does not fully meet PLOS ONE’s publication criteria as it currently stands. Therefore, we invite you to submit a revised version of the manuscript that addresses the points raised during the review process.

The attached review is comprehensive, please make edits in respond to each reviewer's remark

We look forward to receiving your revised manuscript.

Kind regards,

Itzhak Benenson, Ph.D.

Academic Editor

PLOS ONE

Journal Requirements:

This research is supported by National Science Foundation “RAPID: An Ensemble Approach to Combine Predictions from COVID-19 Simulations” grant DEB-2030685.We note that you have provided funding information that is not currently declared in your Funding Statement. However, funding information should not appear in the Acknowledgments section or other areas of your manuscript. We will only publish funding information present in the Funding Statement section of the online submission form.

T.A. A.Z. H.K.

Award no: 2030685

Funder: National Science Foundation

Funder website: nsf.gov

3. We note that Figures 5, 6, 7, 9, 10 in your submission contain map images which may be copyrighted. All PLOS content is published under the Creative Commons Attribution License (CC BY 4.0), which means that the manuscript, images, and Supporting Information files will be freely available online, and any third party is permitted to access, download, copy, distribute, and use these materials in any way, even commercially, with proper attribution. For these reasons, we cannot publish previously copyrighted maps or satellite images created using proprietary data, such as Google software (Google Maps, Street View, and Earth). For more information, see our copyright guidelines: http://journals.plos.org/plosone/s/licenses-and-copyright.

3a, You may seek permission from the original copyright holder of Figures 5, 6, 7, 9, 10 to publish the content specifically under the CC BY 4.0 license. 

3b, If you are unable to obtain permission from the original copyright holder to publish these figures under the CC BY 4.0 license or if the copyright holder’s requirements are incompatible with the CC BY 4.0 license, please either i) remove the figure or ii) supply a replacement figure that complies with the CC BY 4.0 license. Please check copyright information on all replacement figures and update the figure caption with source information. If applicable, please specify in the figure caption text when a figure is similar but not identical to the original image and is therefore for illustrative purposes only.

Reviewers' comments:

Reviewer's Responses to Questions

**Comments to the Author**

1. Is the manuscript technically sound, and do the data support the conclusions?

Reviewer #1: Yes

2. Has the statistical analysis been performed appropriately and rigorously? 

Reviewer #1: Yes

3. Have the authors made all data underlying the findings in their manuscript fully available?

Reviewer #1: Yes

4. Is the manuscript presented in an intelligible fashion and written in standard English?

Reviewer #1: No

5. Review Comments to the Author

Reviewer #1: This is a well-constructed and useful paper. The authors present a robust methodology, which is essentially a large scale clustering project, and with which I have no quibbles. My comments pertain more to reconciliation of data science vs. epidemiology terminology, and the need to further caveat the data sources.

Abstract: "Mobility in 2020 for US counties can be explained as a combination of three trends"

It might be helpful to state these in the abstract.

Table 1: This is excellent and useful. I have made a nearly identical table in my own research, so glad to see this find its way into print.

Methods section 2: "Delta Measure of Public Exposure"

This is semantic: I understand the meaning of this within this context. However, as an epidemiologist, the term "exposure" has a more specific meaning, which is not adequately represented by gross mobility measures. Within epidemiology, "exposure" means direct exposure to the causative agent. Mobility is a proxy for that. The authors may consider modifying the term to avoid detracting from what it otherwise and excellent paper. Similarly, in the discussion "Long term exposure reduction" will likely raise the hackles of public health audiences. Consider "Long term mobility restriction" as a more accurate representation of what was measured.

Discussion: While the data providers have gone to unprecedented lengths to make these data public during the pandemic, as the authors note, the descriptions of methods are sparse. In addition, there are few safeguards that the data will continue to be available, or any real measures of completeness. As such, more in the discussion is warranted about the potential limitations scientific reliance on corporate free data.

2.2 "We note that this measure only includes public POIs that are captured among the 6.5 million POIs in the SafeGraph Core Places database."

This is a perfect example of my previous point. We have no idea how dynamic or complete these POI are. Granted SafeGraph has a more transparent and vibrant community of practice than most of the other providers. But still, the reliance on proprietary data has limitations that could be more clearly addressed.

Methods: Please describe geographic missingness in the dataset, e.g., arising from fewer POI or lower cell trace volume, or measurement uncertainty. What percent of US counties are covered?

"To reduce the dimensionality of R we truncate the SVD to obtain only the first K dimensions "

Please define kappa a little better to give a real world sense of the truncation.

"SVD assumes that the ∆MoPE is derived from the sum of latent (individual) mobility changes."

Is it sum or averaged? The mobility measures usually report mean changes, so there is an additional assumption there.

Methods: How are the 3 latent PCs measured/differentiated in the aggregate mobility data? For example, how are health professionals identified?

Methods: Can you provide more information on SafeGraph "footfall" metric? Does this correspond more in urban areas where foot traffic is more common?

Methods: "it was discovered that the only neighboring county to Fairfax City in Fairfax County,"

Virginia is an annoying case in county-city geography, in my experience. Were both city and county designations used?

Methods: "Pearson’s R coefficient "

This is a measure of linear association. Is there any a priori reason to believe the relationship to be linear?

Methods: "variety of explanatory variables, including income, political leaning, employment, and COVID-19 cases and deaths for each county "

Is this the complete list of variables? COVID cases and deaths were not consistently reported during the study period. For example, were antibody and rapid test positives included? Was repeat testing for the same individual accounted for? Despite being widely reported in data science and news media, the epidemiologic veracity of case counts is highly questionable. This should be caveated accordingly. Other journals have refused to publish results based solely on these numbers.

Figure 3 vs. 4 -- Suggest a different color scheme between graphs because causal readers may confuse the two legends since the lines are so similar.

Figure 6: I do not understand the color wheel and the colored lines extended from it?

I am curious why baseline commute times/distances, weather, and stay-at-home orders were not considered as explanatory variables?

The figure resolution and map projection was of low quality in the review PDF. I assume this will be corrected in the final version.

6. PLOS authors have the option to publish the peer review history of their article (what does this mean?). If published, this will include your full peer review and any attached files.

Reviewer #1: **Yes: **Nabarun Dasgupta

---

## [Author Response · Author response to Decision Letter 0]

16 Jun 2021

Manuscript ID PONE-D-21-10041

Change of human mobility during COVID-19: A United States case study

PLOSONE

We are grateful to the Editor and to the anonymous reviewer for their feedback. We have addressed all comments and indicated where in the manuscript the changes have been made, marked in blue and red font. 

Our responses to the comments are as follows:

Response to Editor Comments:

The manuscript meets PLOS ONE’s style requirements.

This research is supported by National Science Foundation “RAPID: An Ensemble Approach to Combine Predictions from COVID-19 Simulations” grant DEB-2030685.We note that you have provided funding information that is not currently declared in your Funding Statement. However, funding information should not appear in the Acknowledgments section or other areas of your manuscript. We will only publish funding information present in the Funding Statement section of the online submission form.

T.A. A.Z. H.K.

Award no: 2030685

Funder: National Science Foundation

Funder website: nsf.gov

There are two awards that should be acknowledged. Please update the funding statement as follows:

Award 1:

T.A. A.Z. H.K.

Award no: 2030685

Funder: National Science Foundation

Funder website: nsf.gov

Award 2:

T.A. A.Z. H.K.

Award no: 2109647

Funder: National Science Foundation

Funder website: nsf.gov

 3. We note that Figures 5, 6, 7, 9, 10 in your submission contain map images which may be copyrighted. All PLOS content is published under the Creative Commons Attribution License (CC BY 4.0), which means that the manuscript, images, and Supporting Information files will be freely available online, and any third party is permitted to access, download, copy, distribute, and use these materials in any way, even commercially, with proper attribution. For these reasons, we cannot publish previously copyrighted maps or satellite images created using proprietary data, such as Google software (Google Maps, Street View, and Earth). For more information, see our copyright guidelines: http://journals.plos.org/plosone/s/licenses-and-copyright.

We can confirm that the figures that contain map images do not contain copyrighted material.

We have ensured that the reference list is complete and correct.

Response to Reviewers Comments

1. Is the manuscript technically sound, and does the data support the conclusions?

Reviewer #1: Yes

2. Has the statistical analysis been performed appropriately and rigorously?

Reviewer #1: Yes

3. Have the authors made all data underlying the findings in their manuscript fully available?

Reviewer #1: Yes

4. Is the manuscript presented in an intelligible fashion and written in standard English?

Reviewer #1: No

We have revised the manuscript and corrected for any typographical and grammatical errors.

5. Review Comments to the Author

Reviewer #1: This is a well-constructed and useful paper. The authors present a robust methodology, which is essentially a large scale clustering project, and with which I have no quibbles. My comments pertain more to reconciliation of data science vs. epidemiology terminology, and the need to further caveat the data sources.

Thank you for your feedback. Please note we have itemized each comment 5.1., 5.2., and so on to help in the organization of our responses.

5.1. Abstract: "Mobility in 2020 for US counties can be explained as a combination of three trends". It might be helpful to state these in the abstract.

We have added a brief description of the three trends in the abstract.

5.2. Table 1: This is excellent and useful. I have made a nearly identical table in my own research, so glad to see this find its way into print.

Thank you very much.

5.3. Methods section 2: "Delta Measure of Public Exposure"

This is semantic: I understand the meaning of this within this context. However, as an epidemiologist, the term "exposure" has a more specific meaning, which is not adequately represented by gross mobility measures. Within epidemiology, "exposure" means direct exposure to the causative agent. Mobility is a proxy for that. The authors may consider modifying the term to avoid detracting from what it otherwise and excellent paper. Similarly, in the discussion "Long term exposure reduction" will likely raise the hackles of public health audiences. Consider "Long term mobility restriction" as a more accurate representation of what was measured.

Thank you for this feedback. We have renamed the Measure of Public Exposure (MoPE) and ΔMoPE to Time Spent in Public Places (TSPP) and ΔTSPP to more accurately reflect what was measured. We consider an increase in ΔTSPP as one proxy for increase in mobility, thus increasing risk of exposure. We have revised all parts of the paper to consistently reflect this change.

5.4. Discussion: While the data providers have gone to unprecedented lengths to make these data public during the pandemic, as the authors note, the descriptions of methods are sparse. 

We have added additional details in Section 2.1., page 4 regarding SafeGraph’s methods for data collection. We have added a discussion regarding Safegraph’s sparse documentation for their methods for collecting and processing the data in Section 4, page 12. See also our response to comment 5.6 and 5.7.

5.5. In addition, there are few safeguards that the data will continue to be available, or any real measures of completeness. As such, more in the discussion is warranted about the potential limitations scientific reliance on corporate free data.

As part of the SafeGraph Data for Academics, academic researchers have no-cost access to SafeGraph data for non-commercial work. We agree that there are no safeguards, since changes to data access may occur beyond our control. For example, Safegraph recently stopped updating the Social Distancing Metrics dataset. However, the archived data is still available for academic researchers so that the median_non_home_dwell_time can be obtained from 01/01/2019 through 12/31/2020, ensuring the reproducibility of this study. We have included step-by-step instructions and the Python scripts for obtaining and transforming the data from Safegraph’s median_non_home_dwell_time to the ΔTSPP on our github (https://github.com/GMU-GGS-NSF-ABM-Research/Mobility-Trends). We have added these details in Section 2.1., page 4 and Section 4, page 13.

To overcome the potential issues with changes to data access, we have made our derived data, ΔTSPP (for each county and each day), fully available on the GitHub repository. We have clarified this in the Supplemental Material, Section 2.

We have added a discussion on data access and elaborated on the issues of relying on corporate data.

5.6. [Section] 2.2 "We note that this measure only includes public POIs that are captured among the 6.5 million POIs in the SafeGraph Core Places database."

This is a perfect example of my previous point. We have no idea how dynamic or complete these POI are. Granted SafeGraph has a more transparent and vibrant community of practice than most of the other providers. But still, the reliance on proprietary data has limitations that could be more clearly addressed.

We have added additional details in Section 4, page 12 and 13 to quantify the completeness and bias of the data. See also our response to comment 5.4 and 5.7.

5.7. Methods: Please describe geographic missingness in the dataset, e.g., arising from fewer POI or lower cell trace volume, or measurement uncertainty. What percent of US counties are covered?

We have performed additional analysis to quantify the geographic completeness of the data. We have added these details to the Discussion on Section 4, page 12 and 13 and Supplemental Materials. Specifically, we had added Figure 2 to the Supplementary Material to See also our response to comment 5.4 and 5.6.

5.8. "To reduce the dimensionality of R we truncate the SVD to obtain only the first K dimensions "

Please define kappa a little better to give a real world sense of the truncation.

In PCA, k is the number of latent features we wish to retain, where in our study k = 3 (PC1, PC2, PC3). There was a gap in the text where we switched abruptly from k to 3 without explaining. We have clarified this in the text in Section 2.3, page 5. 

5.9. "SVD assumes that the ∆MoPE is derived from the sum of latent (individual) mobility changes."

Is it sum or averaged? The mobility measures usually report mean changes, so there is an additional assumption there.

We have corrected this on Section 2.3, page 5. SVD assumes that the true mobility is the linear combination of latent features which represent mobility change.

5.10. Methods: How are the 3 latent PCs measured/differentiated in the aggregate mobility data? For example, how are health professionals identified?

We make assumptions based on each latent feature or mobility pattern about the individuals that might contribute to these patterns. We have clarified this in Section 3.2.1., page 8.

5.11. Methods: Can you provide more information on SafeGraph "footfall" metric? Does this correspond more in urban areas where foot traffic is more common?

SafeGraph identifies when a device visits a POI using a visit attribution algorithm (https://www.safegraph.com/blog/revealing-safegraphs-secret-method-for-getting-accurate-store-visits-from-gps-data). Foot traffic or footfall in this respect refers to setting foot in a POI rather than walking on foot to a POI.

5.12. Methods: "it was discovered that the only neighboring county to Fairfax City in Fairfax County,"

Virginia is an annoying case in county-city geography, in my experience. Were both city and county designations used?

Anything that has a FIPs code attached to it (counties, cities, and boroughs) is designated a county. Thus, Fairfax City is identified at the county level. This has been clarified in the text in Section 2, page 3.

5.13. Methods: "Pearson’s R coefficient "

This is a measure of linear association. Is there any a priori reason to believe the relationship to be linear?

Our hypothesis is that there is a linear relationship between the strength of the PCs in explaining county mobility and different county variables. We did not investigate non-linear relationships. We clarify this in Section 2.6, page 6.

5.14. Methods: "variety of explanatory variables, including income, political leaning, employment, and COVID-19 cases and deaths for each county "

Is this the complete list of variables? COVID cases and deaths were not consistently reported during the study period. For example, were antibody and rapid test positives included? Was repeat testing for the same individual accounted for? Despite being widely reported in data science and news media, the epidemiologic veracity of case counts is highly questionable. This should be caveated accordingly. Other journals have refused to publish results based solely on these numbers.

The complete list of variables can be found in Table 3. We have acknowledged the caveat of using COVID-19 cases and deaths in Section 3.5, page 11.

5.15. Figure 3 vs. 4 -- Suggest a different color scheme between graphs because causal readers may confuse the two legends since the lines are so similar.

We have changed the colors of Figure 3.

5.16. Figure 6: I do not understand the color wheel and the colored lines extended from it?

We have corrected Figure 6 so that the color wheel and the colored lines match.

5.17. I am curious why baseline commute times/distances, weather, and stay-at-home orders were not considered as explanatory variables?

Thank you. We will consider this in future work. We have added this to Section 4, page 13.

5.18. The figure resolution and map projection was of low quality in the review PDF. I assume this will be corrected in the final version.

We confirm that we have indeed submitted high resolution figures.

---

## [Decision Letter · Decision Letter 1]

19 Aug 2021

PONE-D-21-10041R1

Change of human mobility during COVID-19: A United States case study

PLOS ONE

Dear Dr. Anderson,

Thank you for submitting your manuscript to PLOS ONE. After careful consideration, we feel that it has merit but does not fully meet PLOS ONE’s publication criteria as it currently stands. Therefore, we invite you to submit a revised version of the manuscript that addresses the points raised during the review process.

Academic editor: Reviewer #2 raises important methodological remarks, please react to these remarks.

We look forward to receiving your revised manuscript.

Kind regards,

Itzhak Benenson, Ph.D.

Academic Editor

PLOS ONE

Reviewers' comments:

Reviewer's Responses to Questions

**Comments to the Author**

1. If the authors have adequately addressed your comments raised in a previous round of review and you feel that this manuscript is now acceptable for publication, you may indicate that here to bypass the “Comments to the Author” section, enter your conflict of interest statement in the “Confidential to Editor” section, and submit your "Accept" recommendation.

Reviewer #1: All comments have been addressed

Reviewer #2: (No Response)

2. Is the manuscript technically sound, and do the data support the conclusions?

Reviewer #1: Yes

Reviewer #2: Yes

3. Has the statistical analysis been performed appropriately and rigorously? 

Reviewer #1: Yes

Reviewer #2: No

4. Have the authors made all data underlying the findings in their manuscript fully available?

Reviewer #1: Yes

Reviewer #2: No

5. Is the manuscript presented in an intelligible fashion and written in standard English?

Reviewer #1: Yes

Reviewer #2: Yes

6. Review Comments to the Author

Reviewer #1: The edits and revision look good. Thanks for the attention to detail! The changes, both semantic and substantive are adequate. This is a nice paper. I look forward to being able to cite it.

Reviewer #2: I have read the manuscript “Change of human mobility during COVID-19: A United States case study”. The manuscript suggest a novel approach to identify the changes in human mobility between the years 2019 and 2020, using SafeGraph measure of median non home dwell time for each US county. The authors estimate the daily change in the measure between 2019 and 2020 in each county, find the principal components that explain most of the variance in the received time series, cluster the different county according to their principal components, conduct a spatial autocorrelation analysis between the counties, and a general correlation analysis of socio-demographic variables. The manuscript is concise, clear, and presents a solid framework for analyzing the change in mobility across the US due to the COVID-19 pandemic. However, the manuscript includes major pitfalls that should be addressed to in order to be published.

I divide my comments into general comments and specific comments.

General

1. In figure 2, it seems that TSPP in the summer of 2019 is lower than TSPP in January 2020. Although the authors do relate to the issue of the seasonal anomalies in the winters of both years, it seems highly unlikely that TSPP is higher in January of 2020 than in the summer of 2019. I suggest the authors run another check over the calculation to observe any mistakes. However, if no mistakes are found, I suggest the authors provide accurate data regarding the weather in each county and show that the weather in each county was indeed extreme. Since county s are not equal in size, a bias towards more dense areas exists, and extreme weather conditions in one of the metropolitan areas can affect the results. However, due to the fact that no data from previous years exist, and 2019 is presumed to be a baseline year, it is crucial to validate the weather conditions(for example, by using NOAA’s GHCN data) against the TSPP in order to explain the anomalies in TSPP.

2. Regarding the principle components analysis – what led the decision to choose only 3 PCs? It is important to show the distribution of the contribution of the PCs to the variability, and then explain what led to the decision to choose only the first three.

3. The choice of k = 3 in k means clustering cannot be explained using the 3 significant PCs. I suggest the author find a better explanation for using k =3, such as the k means bend method or other methods that exist for hierarchical clustering.

Specific

1. In definition 1, j should run between -3 and 3, in order to should that the moving average is of the middle day of the 7 day average

2. In definition 2, it should be noted that delta-TSPP is a by county measure, and therefore it should be parametrized and indexed as well.

3. In figure 2, an error bar/boxplot should be added for each day, in order to get a sense of the distribution of TSPP in the different counties.

4. 2.5 page 6: what state is Fairfax county?

5. Figure 6 is unreadable. I would remove it from the paper as it confuses the reader, and it is somewhat redundant given figure 5.

6. I suggest adding the share of population over 60 as one of the correlation analysis socio-demographic variables – it can help explain the positive coefficient of deaths per thousand in PC1, which is counterintuitive.

7. PLOS authors have the option to publish the peer review history of their article (what does this mean?). If published, this will include your full peer review and any attached files.

Reviewer #1: **Yes: **Nabarun Dasgupta, MPH, PhD

Reviewer #2: No

---

## [Author Response · Author response to Decision Letter 1]

7 Oct 2021

Manuscript ID PONE-D-21-10041R1

Change of human mobility during COVID-19: A United States case study

PLOSONE

We are grateful to the Editor and to the two reviewers for their feedback. We have addressed all comments and indicated where in the manuscript the changes have been made, marked in blue and red font. 

Our responses to the comments are as follows (>>):

Reviewer #1 

The edits and revision look good. Thanks for the attention to detail! The changes, both semantic and substantive are adequate. This is a nice paper. I look forward to being able to cite it.

>>Thank you.

Reviewer #2 

I have read the manuscript “Change of human mobility during COVID-19: A United States case study”. The manuscript suggest a novel approach to identify the changes in human mobility between the years 2019 and 2020, using SafeGraph measure of median non home dwell time for each US county. The authors estimate the daily change in the measure between 2019 and 2020 in each county, find the principal components that explain most of the variance in the received time series, cluster the different county according to their principal components, conduct a spatial autocorrelation analysis between the counties, and a general correlation analysis of socio-demographic variables. The manuscript is concise, clear, and presents a solid framework for analyzing the change in mobility across the US due to the COVID-19 pandemic. However, the manuscript includes major pitfalls that should be addressed to in order to be published.

I divide my comments into general comments and specific comments.

General

In figure 2, it seems that TSPP in the summer of 2019 is lower than TSPP in January 2020. Although the authors do relate to the issue of the seasonal anomalies in the winters of both years, it seems highly unlikely that TSPP is higher in January of 2020 than in the summer of 2019. I suggest the authors run another check over the calculation to observe any mistakes. However, if no mistakes are found, I suggest the authors provide accurate data regarding the weather in each county and show that the weather in each county was indeed extreme. Since county s are not equal in size, a bias towards more dense areas exists, and extreme weather conditions in one of the metropolitan areas can affect the results. However, due to the fact that no data from previous years exist, and 2019 is presumed to be a baseline year, it is crucial to validate the weather conditions(for example, by using NOAA’s GHCN data) against the TSPP in order to explain the anomalies in TSPP.

>>Based on NOAA’s GHCN data, it is clear that the US as a whole experienced a mild winter season (January - April 2020) with warmer average temperatures and roughly 50% less snow on the ground than in comparison to the 2019 winter season. This, in combination with panic buying behavior at the beginning of the pandemic, helps to explain the high mobility in January, February, and early March. We have updated the text and added Figure 3 (snow depth) and 4 (average temperature) to Section 3.1 of the Supplemental Materials to present these findings. Any further analysis to explain the mobility data based on weather conditions goes beyond the objective and scope of the study.

Regarding the principle components analysis – what led the decision to choose only 3 PCs? It is important to show the distribution of the contribution of the PCs to the variability, and then explain what led to the decision to choose only the first three.

>>The explained variance for each PC is presented in Figure 5 and Table 1 of Section 3.3. of the Supplemental Materials. We use the elbow method to determine that 3 PCs are sufficient. 

The choice of k = 3 in k means clustering cannot be explained using the 3 significant PCs. I suggest the author find a better explanation for using k =3, such as the k means bend method or other methods that exist for hierarchical clustering.

>>We use the k-means clustering as a visualization tool so that we can observe which counties have a similar weighted combination of the three PCs. Thus, we arbitrarily choose a value of k = 3. Although we find that the silhouette method finds a better explanation where k = 6 (see Figure 6 in Section 3.4 of the Supplementary Materials), it is not as effective visually (see Figure 7 in the Supplementary Materials).

Specific

In definition 1, j should run between -3 and 3, in order to should that the moving average is of the middle day of the 7 day average

>>The reviewer is correct. In our implementation we indeed use a centered moving average of seven days. Definition 1 has been corrected to for the index j to run between -3 and 3, and we have clarified that the defined TSPP of a time series is only defined between the fourth and fourth-to-last element of the time series.

In definition 2, it should be noted that delta-TSPP is a by county measure, and therefore it should be parametrized and indexed as well.

>>The parameter R denotes the spatial region, such as a county. This was already parameterized in the previous manuscript. No changes were made to Definition 2.

In figure 2, an error bar/boxplot should be added for each day, in order to get a sense of the distribution of TSPP in the different counties.’

>>We have added Figure 1 and Figure 2, presenting the boxplots for each week to the Supplementary Materials.

2.5 page 6: what state is Fairfax county?

>>We have updated the text with this information to clarify that Fairfax County is part of Virginia.

Figure 6 is unreadable. I would remove it from the paper as it confuses the reader, and it is somewhat redundant given figure 5.

>>We have moved this figure to Section 3.5 of the Supplemental Materials.

I suggest adding the share of population over 60 as one of the correlation analysis socio-demographic variables – it can help explain the positive coefficient of deaths per thousand in PC1, which is counterintuitive.

>>We have added the percent population over 65 as one of the correlation variables.

---

## [Editor Report · Decision Letter 2]

12 Oct 2021

Change of human mobility during COVID-19: A United States case study

PONE-D-21-10041R2

Dear Dr. Anderson,

We’re pleased to inform you that your manuscript has been judged scientifically suitable for publication and will be formally accepted for publication once it meets all outstanding technical requirements.

Kind regards,

Itzhak Benenson, Ph.D.

Academic Editor

PLOS ONE
---

## [Editor Report · Acceptance letter]

25 Oct 2021

PONE-D-21-10041R2 

Change of human mobility during COVID-19: A United States case study 

Dear Dr. Anderson:

I'm pleased to inform you that your manuscript has been deemed suitable for publication in PLOS ONE. Congratulations! Your manuscript is now with our production department. 

Kind regards, 

on behalf of

Professor Itzhak Benenson 

Academic Editor

PLOS ONE